# Revealing Unreported Benefits of Digital Water Metering: Literature Review and Expert Opinions

**Ian Monks [1,2], Rodney A. Stewart [1,2,\*], Oz Sahin [1,2] and Robert Keller [3,4]**

[1] School of Engineering and Built Environment, Griffith University, Gold Coast 4222, Queensland, Australia; ian.monks@griffithuni.edu.au (I.M.); o.sahin@griffith.edu.au (O.S.)

[2] Cities Research Institute, Griffith University, Gold Coast 4222, Queensland, Australia

[3] Engineering Department, Monash University, Clayton 3800, Victoria, Australia; rjkeller@optusnet.com.au

[4] R. J. Keller & Associates, Wheelers Hill 3150, Victoria, Australia

\* Correspondence: r.stewart@griffith.edu.au

**Abstract:** Digital water meters can take Australian water utilities into the world of internet of things (IoT) and big data analytics. The potential is there for them to build more efficient processes, to enable new products and services to be offered, to defer expensive capital works, and for water conservation to be achieved. However, utilities are not mounting business cases with sufficient benefits to cover the project and operational costs. This study undertakes a literature review and interviews of industry experts in the search for unreported benefits that might be considered for inclusion in business cases. It identifies seventy-five possible benefits of which fifty-seven are classified as benefiting the water utility and forty are classified as benefiting customers (twenty-two benefit both). Many benefits may be difficult to monetize. Benefits to customers may have a small monetary benefit to the water utility but provide a significant benefit to customer satisfaction scores. However, for utilities to achieve these potential benefits, eight change enablers were identified as being required in their systems, processes, and resources. Of the seventy-five benefits, approximately half might be considered previously unreported. Finally, a taxonomy is presented into which the benefits are classified, and the enabling business changes for them to be realized are identified. Water utilities might consider the taxonomy, the benefits, and the changes required to enable the benefits when developing their business cases.

**Keywords:** digital water metering; smart water networks; benefits taxonomy; digital disruption

## 1. Introduction

The purpose of this review is to prepare a comprehensive catalogue of the benefits of digital water metering. Many researchers have studied various aspects of the impacts of digital metering from trials and rollouts areas across the world [1–15]. The situation in Australia mirrors most other advanced economies internationally, where only a small number of water utilities have fully or partially deployed digital water meters (e.g., Mackay Regional Council), with a few more deploying projects in progress or about to start (e.g., TasWater), but the majority are only conducting small-scale digital metering trials.

The slow rollout of citywide digital metering projects is often due to business cases falling short for the value of benefits to cover the costs. Anecdotally, some Australian water utilities describe the technology being considered as immature and that they lack confidence in the benefits being achieved. Boyle et al. reviewed the drivers and challenges of intelligent metering implementations [16]. Mutchek and Williams [17] discussed the sustainability and resilience issues faced by water utilities and the economic disincentives against developing smart water grids. However, groups like

the California Water Association urged authorities to implement Advanced Metering Infrastructure (AMI) technology in the face of crippling drought [18].

The business cases that have been approved have sufficiently high-value compelling drivers such as deferred network augmentation [19], water conservation [1], or customer service [20]. In their study of digital metering in Alicante, Spain, March et al. [6] commented on the struggle that water managers may have in identifying all the benefits at the proposal stage and determining the monetary value of the benefits. In compiling this list of benefits, the authors seek to assist managers in identifying benefits relevant to their operation and appetite for change. No quantification of the benefits is attempted in this paper.

Successful deployments have been documented for a number of cities and regions. While the benefits and, in some cases, the process are described [6,20–24], the quantum of the benefits obtained is often not included. It is often left to the suppliers of the systems to publish the outcomes rather than the water companies.

In Victoria, Australia, the conservative approach to digital metering adoption within water utilities might be traced back to the compulsory rollout of electricity smart meters. Customers were required to fund the changeover despite the lack of benefits delivered to them while electricity businesses reaped the benefits. The Energy and Water Ombudsman of Victoria (EWOV) reported on the complaint cases that they received from the rollout [25]. Messner, in his book on preparing business cases, included the Victorian electricity smart meter rollout as a case study of a failed project [26]. Electricity customers' resentment surfaced again in 2018 when it was suggested that they would be asked to pay for an upgrade to the electricity meters [27]. Also, in the 2018 pricing submissions to Victoria's Essential Services Commission (ESC), the Consumer Action Law Centre (CALC) reminded the commission and corporations that "smart energy meters were touted as a game changer…but have so far failed to deliver on this promise" and "that new technology must deliver tangible benefits for water customers and be backed by a comprehensive business case" [28].

Beal and Flynn conducted a digital utility survey and recommended that the planning and vision for how a digital water network can work for a utility is just as important as the technology itself [1].

The cost side of a digital metering rollout is subject to the technology selected, the extent of the implementation, and the purchasing processes of individual water utilities. Assessing rollout costs are outside the scope of this present paper.

## 2. Study Methods

With so many water utilities struggling to find sufficient benefits to include in cost–benefit analyses and business cases for the rollout of digital water meters, this literature review and interviews of industry experts sought to identify less obvious, hidden, and intangible benefits that might be considered for inclusion in proposals.

### 2.1. Literature Review

The literature review searched for journal and conference papers describing case studies and reviews of digital metering deployments and trials. From the websites of water utilities annual reports for a number of years, water plans and other documents relevant to customer services, metering, and hardship programs were downloaded and reviewed. The purpose was to understand their business situation and to identify operational issues where digital metering might provide some benefit.

Moreover, a number of water utilities from across Australia and some related organisations consented to participation in the research. These participating water utilities were able to provide a number of internal, unpublished documents of metering trials, studies, and proposals on a "commercial in confidence" basis. In 2017, the Victorian water utilities were required by the ESC to consult customers and to prepare their 2018 pricing submissions. These submissions were downloaded [29] along with the customer comments and the ESC reviews and determinations. In these documents, the water utilities reported their progress towards digital water metering.

This literature review identifies many well-covered benefits and other potential benefits not broadly covered in case studies. To reinforce the validity of these benefits in the Australian situation, the opinions of experts were also sought.

### 2.2. Expert Opinions

Expert opinions were solicited using a series of unstructured and structured interviews as well as literature reinforcement. Five unstructured interviews were conducted with industry experts from a cross section of disciplines and with extensive and broad experience and one from outside the industry (insurance assessor). The purpose was to fine-tune the benefits in preparation for putting them to a larger representative group of current and recent water professionals. As a result, seventy-two benefits were synthesized into a structured interview along with some background questions to profile the participants. Based on their feedback, a further round of literature searching was undertaken and a second structured interview was undertaken with selected participants.

In total, fifty-two experts completed the first structured interview. They represented a broad range of disciplines, years in the industry, and levels of experience with digital metering. Figure 1 illustrates some descriptive statistics for the interviewed participants of the first structured interview (Figure 1). The experience and expertise of these participants are listed in Table A1 (see Appendix A).

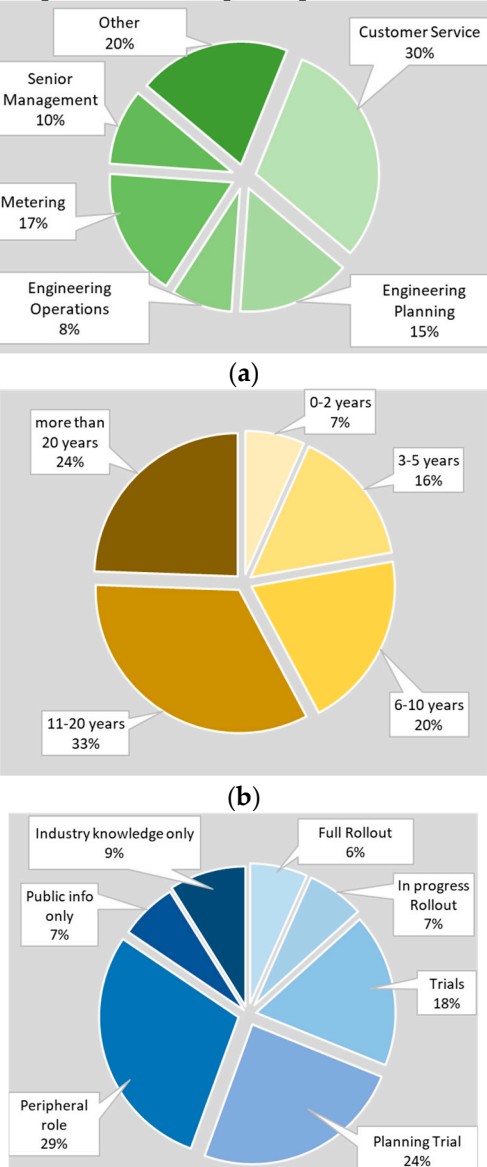

**Figure 1.** Profiles of first structured interview respondents: (**a**) The business area experience of interviewed water-industry staff; (**b**) the years of water-industry experience; and (**c**) the levels of knowledge of digital water metering through rollouts, trials, and public information.

From the structured interviews and a subsequent seventh unstructured interview (Owners Corporation Manager), a further three benefits were identified, bringing the total to seventy-five. These benefits are catalogued in the following sections.

## 3. Enabling the Achievement of Benefits

Digital water metering can use Automated Meter Reading (AMR) technology, where local devices on mechanical meters are read by walk-by handheld readers or drive-by readers in vehicles. The data is offloaded from the device into a central system back at the office. The interval of the data recorded is determined by the period between walk-by's or drive-by's. AMI technology adds a communication layer that reads the meter automatically on fixed intervals, usually hourly, and transmits the data back to the central system [1,8]. The more complex the technology, the greater the cost to establish and operate. Hourly read data produces around 2200 readings per quarter against the single value recorded by manual meter reading. Normal billing function takes the digital reading at a nominated date and time of the billing cycle for the bill. The large, constantly growing dataset opens up the world of big data to water companies that might then undertake data mining to better understand their business, leading to more informed planning and operations, cost savings, and new products and services. Allowing customers to view their data directly can lead to more informed conversations and reduce the number of direct contacts as customers self-service their information enquiries. While water companies have operated SCADA systems for a long time to monitor critical functions across their network, digital metering at the customer level takes them deep into the modern world of the internet of things (IoT), digital disruption, and data analytics [3].

Digital water metering alone will not deliver the benefits. The water utilities also need to change their systems, processes, and resourcing and offer new services [11,20,30,31]. Eight primary enabling requirements have been identified (Table 1).

**Table 1.** The changes required to enable various benefits to be achieved.

| Enabler | Water Industry Role(s) | References |
|:---:|:---:|:---:|
| 1 | Automate meter reading using advanced meter infrastructure | [1,6,17,20,32] |
| 2 | Improve demand and revenue forecasting through advanced data analytics | [11] |
| 3 | Establish leak alerting | [20,23,33] |
| 4 | Establish a detailed customer data portal for single and complex properties | [5,6,23,34,35] |
| 5 | Offer monthly billing | [1,36] |
| 6 | Establish detailed water balancing of permanent and temporary DMAs | [23,35,37,38] |
| 7 | Establish a capability for meter/metering/end-use data analytics | [6,11,39] |
| 8 | Increase knowledge of customers and assets | [9,40,41] |

Note: The enabling changes required for each benefit are identified in the Benefit Catalogues of Tables 2–4 in the Sections 4 (Business Benefits), 5 (Customer Benefits), and 6 (Shared Benefits), respectively.

### 3.1. Enabler#1—Automate Meter Reading using Advanced Meter Infrastructure

Water meters fitted with digital counters and communications devices can be read by walk-by readers and are being read this way for customers with hard-to-access meters [1]. An alternative to walk-by is drive-by reading [17,32]. However, these only provide one reading per billing period or drive-by.

A fundamental benefit of digital water meters is the enablement of more frequent automated reading through a communications network (AMI) and the flow on cost saving from avoiding staff walking around taking manual meter readings. The reading is date–time stamped and automatically transferred to a central database from where it can be read by the various systems—billing, customer portal, customer call centre, and data analytics [6].

Not all manual meter reading is eliminated as meter replacement requires the old and new meter details to be read and input to the central database. Additionally, the accuracy of the system should be verified through field audits, occasionally. The paper by Thiemann et al. describes how AMI enabled Kansas City's business processes to be streamlined to reduce field trips [20].

### 3.2. Enabler#2—Improve Demand and Revenue Forecasting through Advanced Data Analytics

A second fundamental benefit is that the mass of data, twenty-four water use readings per day from every customer can be stored and used along with other data such as billing rates, demographic data, and weather forecasts [11]. From these historical records, more accurate demand forecasts can be developed under different weather and customer growth scenarios. Using daily consumption readings, the billable amount can be projected or forecast and more accurate revenue forecasts can be established.

### 3.3. Enabler#3—Establish Leak Alerting

Concealed leaks and other unexplained high water use at properties lead to high water bills as well as wasted water and are an expense to the water utilities when allowances to the customer are given [33].

Relatively simple algorithms running over the 24 daily readings for each customer can identify multiple hours of continuous nonzero use within a property in a day. These algorithms are looking for the minimum night flows (although they may not occur at night for some customers). If a customer's minimum night flow is above zero for one or more days, then it is possible that there is a leak at their property or their water use mimics a leak such as when filling a swimming pool, undertaking night irrigation, or using an evaporative cooler [23].

A fundamental benefit from the use of digital water metering is the automated alerting of the customer to a possible leak. During digital metering trials, the checking and customer notification may be manually undertaken by a Customer Service Representative (CSR) [20,35], or if an automated system is in place, an email, text message, or letter may be sent out. Customer portals such as myh2o (MiWater add-on) have this automated alerting and also have more sophisticated customer-driven minimum night flow settings that must be exceeded before the alerts are sent [23].

In the event that a property is vacant, alerting systems can be configured to alert the property owner to any water use which may indicate water theft, intruders or squatters, or a leak/burst [20].

### 3.4. Enabler#4—Establish a Detailed Customer Data Portal for Single and Complex Properties

The collection and storage of hourly readings of water usage and the internet combine to provide an opportunity to establish customer access to their consumption data in near real-time.

A Sydney study of 120 households found the water conservation benefit of providing customer access to their data using the custom-built online My Home Our Water (MHOW) portal to be 4.2% [5].

The Mackay Regional Council built the MiWater application with both back-office and public-facing (customer) web portals for use by their 40,000 customers and made MiWater available commercially to other water utilities [23,35].

Analysing the impact of customer access to an online portal by the City of Sacramento residents over an 18 month period, Schultz et al. identified a 50% reduction in the occurrence of leaks (down from 12% to 6%) and a 34% reduction in the length of time to fix the leak [34].

### 3.5. Enabler#5—Offer Monthly Billing

Enabling monthly billing is different from providing monthly payment plans. With monthly payment plans, there are phone calls between the customer and the water utility, negotiations around the fixed installment amount to be paid per month, and a reconciliation at the end of the period for over- or underpayments. The monthly payment plan is usually restricted to a single period of 12

months. Further negotiations between the customer and the water utility are required for an extension and new rate [42–44].

With monthly billing, the meter reading at a nominated date and time of the month is the basis for the calculation of water use and charging [1]. There is no intervention other than to switch a flag on the customers' records—which, if provided within a customer account portal, could be done by the customer themselves. The required manual activities by the customer and customer service staff are substantially eliminated.

Some water utilities may need to upgrade or replace their customer billing system to accommodate monthly billing based on readings [36].

### 3.6. Enabler#6—Establish Detailed Water Balancing of Permanent and Temporary DMAs

The use of digital water meters provides customer demand data in time frames that match network flow meter data. Netting out the two enables water balancing to be calculated daily [35].

Systems like MiWater provide a back-office facility to enable the network and District Metering Areas (DMAs) to be defined and the water balances to be calculated [23]. The Hydrant Insertion Device developed by Yarra Valley Water might be used to create temporary DMAs across much smaller subareas of the permanent DMAs [38].

Once water balances are determined, the gap between supply (flow meter volume) and demand (total customer volume) identifies the network leak volume. Maintenance teams can then search for and repair the network where it is leaking, thus conserving water [37].

### 3.7. Enabler#7—Establish a Capability for Meter/Metering/End-Use Data Analytics

Data analytics is not a new undertaking among water utilities as they have been required to report internally and externally to the regulatory bodies, such as Victoria's Essential Services Commission and other Government bodies [45]. What may be new is the employment of trained statisticians and data scientists capable of going beyond simple reporting, undertaking the analysis of large integrated datasets using advanced statistical, data mining, and data analytics techniques and of advancing the data maturity of the organization [46–49].

The establishment of a repository of data from digital water meters that cover the water usage for all customers at consistent hourly intervals is a significant step forward in facilitating data analytics of meters, metering, and end-use [6,11]. This is consistent with the view on big data of Dr. Adrian Letchford who is reported in the *The Australian*, observing that, with big data, the whole picture can be seen and not just a small sample [39].

### 3.8. Enabler#8—Increase Knowledge of Customers and Assets

The water end-use and time-of-day patterns are becoming well-known for suburban residential customers [9,50]. However, other segments of water customers are less well-known. Attempts have been made to understand different business types [41] and different precincts [40].

Digital water metering across all customers will enable water use patterns, at least at hourly intervals, across different days of the week and seasons to be revealed, thus providing greater knowledge of all customers [4].

There is the opportunity to collect considerable customer profile and appliance data volunteered by the customer through a digital water meter portal and from additional public datasets.

## 4. Business Benefits

### 4.1. Operational Cost Savings

#### 4.1.1. Meter Reading

Digital water meters have **reduced the staff and labour costs** required to read meters. In Lincoln, Nabraska, six meter readers was replaced by one in a van with a computer covering 80,000 meters

[22]. Six meter readers had been used in Aiken, South Carolina to read 15,000 m during 3 weeks every month. Using drive-by reading, labour savings of $171,000 had been achieved [32] as well as staff cost savings and workers compensation claims from injuries ranging from bee stings to knee operations related to bending and lifting cast iron lids, amounting to $80,000 over 5 years. Other water companies claimed to **reduce manpower for special readings** when the customer at a property changed for final billing, to **reduce the number of estimated readings** when access to the meter was not available [6,21,51], and to reduce fuel and vehicle maintenance costs and $CO_2$ emissions [11], that is, to gain a benefit of **reduction in vehicle energy costs (Green House Gas (GHG) emissions)**. Kansas City claimed a 33 staff reduction from the introduction of its AMR system and the subsequent redesign of processes. They estimated that about 15% of bills had been based on estimated reads, 35% of customer calls had been related to meter reading, and additional resources were expended to defend the accuracy of its billing process [20]. In 2017, Utility#14 had over 2000 estimated reads on a customer base of 60,000 and these were putting a strain on customer relations and additional staff time to manage the billing process. Moreover, in the financial year 2015/16, over 11,000 special reads were ordered by this utility [19].

Other water companies reported an **avoidance of Occupational Health and Safety (OH&S) issues**, similar to Aiken, South Carolina's experience. AMR reduced meter readers being faced with OH&S issues involving dogs, snakes, pits, and angry customers. A water company nominated a near-90% reduction in OH&S claims from metering activities in their business case [19].

Using digital water meters, readings are accurate and results in **reduced meter reading errors** [1,22,32,51]. There is an opportunity to offer monthly billing to customers using the automated reading taken at a nominated day and time [1,52]. This would increase postal charges, but coupling monthly billing with eBilling significantly reduces the extra mailing costs. Further coupling with direct debit improves the potential for on-time payments and reductions in reminders and other debt recovery expenses. Experts identified as E17, E26, E38, and E43 expressed the opinion that digital metering could enable a **reduction in billing and collection costs when monthly billing was coupled with and electronic billing and collection**, while E58 pointed out that many customers will continue to like their bills posted (see Appendix A for Experts).

Boyle et al., in summarising many international deployments of AMR technology, says that they are "typically rationalised in terms of remote access benefits (e.g., reduced labour costs for meter reading, reduced health and safety risks from hard-to-access properties requiring reaching over fences or confronting pet dogs)" [16].

Beal and Flynn, in their 2014 review of smart metering and intelligent water networks in Australia and New Zealand for the Water Services Association of Australia (WSAA), reported that 16% of responding water companies identified reducing meter reading costs and operating costs as the main driver. Another 6% identified OH&S issues and 5% identified improving nonrevenue meter reads [1]. Their summaries of business cases for City West Water, Victoria and Water Corporation, Western Australia documented outcomes for reduced operating costs from 2013 and 2014. In their review, Beal and Flynn built on the Cost Benefit Analysis spreadsheet developed by the WSAA Cost Benefit Analysis spreadsheet for the Metering Program Group in 2012.

### 4.1.2. Financial Management

Digital water metering can **improve cash flow through monthly billing** by accelerating collections and by reducing expenses through other benefits. In the book *Controlling Costs and Cash Flow—Strengthening your business's financial performance* [53], the efficiency of the whole process of billing and receiving prompt payment is noted for effective cash management. In an American Water Works Association (AWWA) Journal Consultant's Corner article, when describing the selection of the Itron AMR system, Victor Mercado, Detroit Water and Sewerage Department's director, is reported as nominating monthly billing for an improved cash flow [21] and as a benefit of their digital metering project. This view was repeated by Quraishi and Siegert in their Field Report article in the AWWA Journal [54]. Hastreiter reported that Aiken, South Carolina estimated a $400,000 increase in annual revenue as a result of going from self-read to automated reading [32].

Rizzo described automated meter reading as the solution to inaccuracies of manual meter reading and said that meter readers tended to "under-read" meters because it creates fewer problems with the consumer and that there may be an agreement made with the consumer [51]. In their 2014 Review, Beal and Flynn noted that, for the water utilities that had completed a rollout, the benefit that most exceeded expectations was an **improved meter reading accuracy** [1]. Based on these findings, digital water meters can be expected to **reduce residential nonrevenue water data errors/losses** and **to reduce nonresidential nonrevenue water data errors/losses.**

Digital water metering, along with eBilling and direct debit, enables an efficient operation of monthly billing. A further advantage of monthly billing is evidenced by Australian electricity provider Origin Energy who, in their 2016 Annual Report, pointed out that the introduction of monthly billing was a contributor to the **reduction in working capital** (of $365 million) [55].

In their 2011 paper to the Institute of Electronic and Electrical Engineers (IEEE), Li et al. note that smart meters bring a "flood of data" and the need to manage and transform the data into actionable business intelligence [11]. They described how predictive-usage-analysis-based smart meter management and analytics can be applied for **improved revenue forecasting**.

Chesler and Schluman describe the shift from general liability to property policies to cover companies from litigation centred on claims [56]. They describe the possible consequence of infiltration of water into businesses or residential properties as having catastrophic results. Insurance companies offer water utilities insurance to cover general liability claims [57,58]. King and Gage suggest that the IoT is a game changer for managing risk and reducing and predicting claims [59]. It is suggested that earlier network leak detection through the collection and analysis of data from DMAs and digital water meters can **reduce the incidence of insurance claims and costs against water utilities**.

### 4.1.3. Utility Costs

Rizzo pointed out the cost of the water wasted through network leaks [51]. Water conservation through reducing customer leaks, overall demand, and nonrevenue water will reduce the volume of water required to be sourced and, in turn, **reduce the wholesale water cost** to the water company.

In a literature review undertaken in 2017 by Liu and Mukheibir for some of Australia's metropolitan water utilities, the customer water savings from digital water metering and feedback portals ranged from 3–8% (the 10th to 90th percentile envelope, excluding outliers) [60].

Marsh said the water–electricity nexus and the climate change debate had brought the links between water and electricity into focus and added an environmental element to the debate [61]. Candelieri went further and used the data and employed a Support Vector Machine regression to improve the reliability of forecasts and to optimise operations, particularly the pumping schedule [62].

Environmental benefits such as a reduced carbon footprint of water supply systems results from water savings and **reduces the energy required for pumping** during production and distribution. March et al. cites the water company Aguas de Valencia report that had calculated that its smart metering schemes saves about five million cubic meters of water per year and so avoids the emission of 600 tons of carbon dioxide ($CO_2$) [6]. Marsh points out that smart meters monitoring real-time water and electricity use would enforce the link between consumption and bills and that lower consumption reduces electricity consumption [61].

Hourly data capture would provide valuable information on diurnal and peak consumption patterns, enabling better peak demand management including the identification of high users and precincts [1,13].

Li et al. [11] use data analytics to identify theft and to bypass problem by a sudden drop or level shift in usage. In Alicante, Spain, the detection of unauthorised domestic water use (theft) has increased since the introduction of digital water meters in 2011 and increased surveillance by water company employees [63]. They reported an 80% **reduction in water theft** in 2017 from the 2013 figures. Candelieri offers a data analytics method of identifying possible frauds [62].

The early detection of leakage at DMA and local levels is the subject of the Luciani et al. paper on the GST4Water project [64]. Beal and Flynn noted the following comment from a respondent to their survey of digital metering projects: "Timely address costumer issues: Leaks attended to in timely manner once reported to customer—Network leakage reduced once data correlated". Also, in their template for developing a business case of digital metering, Beal and Flynn suggest that there would be less wasted time to detect network leaks [1], that is, a **reduction in network leaks and other non-revenue water (NRW) causes (e.g., bursts)** and a **reduction in labour costs associated with leak detection**. Interviewed expert E17 pointed out that water utilities spend money to understand the source of water loss. Sometimes that turns out to be on the customer's side of the meter, effectively wasting their time as it is the customer's responsibility, but with digital metering, that situation could be identified more quickly. E36 pointed out that customer leak-alerting would also eliminate wasted investigations for network leaks that ultimately are found to be on the customer-side of the meter. Experts also commented on the ability to be more targeted in their search for leaks.

### 4.1.4. Meters

Water meters in Australia are required to operate between accuracy limits of ±4% [65]. While they are accurate early in their life (except for cases of installation failure), they are mechanical devices that wear over time. Without in situ accuracy testing being practical, water companies have policies that require water meters to be replaced after a fixed number of years or whenever the meter reaches a nominated number of kilolitres (kl).

One water utility, Utility#14, was using a replacement criteria of just 7 years or 3000 kl. In an internal report, they pointed to recent research that showed that the technical life of meters was closer to 15 years and that switching the policy would mean limited replacements for 6 years and would defer many millions in capital investment [19]. An analysis of meter testing data at another water company, Utility#10, identified through their data analysis that the life of meters could be in excess of twenty years and for at least another two years before retesting and then for another two years beyond that. [66]. In another study, Utility#13 found similar results [67].

Attempts to use quarterly reads to identify failing meters using data analytics had some success [68]. Digital meters read hourly provide over 2000 more data values over quarterly read meters. To obtain a cost-saving benefit from digital meters from a **deferral of meter replacement** requires a means of differentiating between compliant and noncompliant meters and targeting those meters that are failing so that they can be removed while leaving those meters that continue to comply with the meter accuracy compliance rules. In their paper, titled "Usage Analysis for Smart Meter Management", Li et al. described a predictive usage analytics toolkit used for meter anomaly detection [11]. One cause of meter anomaly was meter failure. They claimed that the toolkit could receive a number of inputs that could be applied to algorithms to detect failing meters. However, the data analytics required need not be advanced. Symmonds demonstrated the use of a simple 30-read running average to identify meter accuracy and component failure [69]. Many of the experts interviewed (E26, E41, E70, and E81) believed that battery life would determine replacement, that is, when the battery is replaced, the whole meter is replaced. However, other Experts (E17, E25, and E31) recognized the potential for data analytics to identify the compliant and failing meters, including E72 who pointed out that the data analytics "should eliminate a lot of assumptions that meter replacement programs are based on" and estimated a 50% reduction.

### 4.1.5. Tariffs

Coliban Water nominated future time-of-use tariffs as a benefit of their rollout project [70]. March et al. reported that, according to company managers in Alicante, remote meter reading might help in designing more **flexible tariff schemes**, either by rethinking tariffs according to consumption patterns or through tariffs differentiating water tariffs according to temporal periods [6]. The electricity industry already uses time-of-use-tariffs (TOUTS) in their pricing structures and in helping to regulate peak demand [16]. Boyle et al. also suggested that intelligent metering offers the same potential for water **load levelling** from peak to off-peak times. Cole and Stewart discussed new tariffs

in response to scarcity and seasons and the potential for penalty charges for exceeding consumption within a one-hour period to manage peak demand [10]. The opinion of experts interviewed (E22, E27, E58, E59, and E61) saw load leveling as a potential future change. Boyle et al. [16] further described the **increased level of customer service and satisfaction** from new tariffs as a societal benefit. Experts' opinions were divided (E19, E22, E38, and E61). However, E59 described flexible tariffs as facilitating the introduction of a smart grid for water.

*4.2. Capital Cost Savings*

4.2.1. Planning

Improved the quality of data input into hydraulic models, especially diurnal patterns using actual hourly consumption data from digital meters for different consumer types, making the models more accurate and efficient [10]. The modelling of areas where there are large commercial customers is improved when the sparse knowledge of their diurnal curves is filled with hourly data. In their 2014 Review, Beal and Flynn reported Mackay Regional Council's new ability to use their data to optimise the design rather than overestimating demand as happens currently. [1]

A number of authors refer to the opportunity provided by so much extra data for understanding temporal patterns under different conditions and for enabling the improvement to infrastructure/network planning [4,11,16]. The sample cost–benefit analysis spreadsheet developed by WSAA includes a line item for savings from **improved network planning** [52].

A reduced consumption and, in particular reducing, consumption on peak days and during peak months would alter the peak consumption parameters when designing infrastructure and sizing of the expensive parts of the network [2,10]; **deferring the augmentation of these assets** would lead to significant financial benefits. A press release by Taggle reported that the Mackay Regional Council was able to defer the building of a new $100M treatment plant from 2022 to 2032 after successfully deploying the Taggle system in January 2013 and running a number of customer engagement initiatives that reduced consumption by 12% [31]. MidCoast Water's interest in measuring the impact on water conservation through a trial of customer water-use information was specifically for the deferral of capital expenditure for supply augmentation [10,71].

4.2.2. Risk

Key obstacles to integrated urban water management include a risk-averse culture with little tolerance for uncertainty [72]. Gurung et al. explains that improved knowledge of end-uses and diurnal patterns will allow network planners to change from using fixed demand estimates and peaking factors to more dynamic approaches based on diurnal patterns to allow more optimal designs [4]. Dr. Adrian Letchford, in arguing for big data and advanced analytics, called into question traditional small cross-sectional sampling and its level of certainty when you can use all of your data [39]. Li et al. [73] described optimised infrastructure planning as a benefit from the use of predictive usage analytics for forecasting at an individual and customer segment level, effectively enabling a **reduced risk premium** in models. The Water Services Association of Australia recognised the potential benefit of **reduced working capital costs** by including it as a line item in its sample cost–benefit analysis spreadsheet [52]. The experts interviewed (E22, E27, E58, and E59) supported the benefit, but some thought it would have a low impact (E19 and E38).

During interviews with water utility staff [36,74], for those water utilities that have not yet done so, the opportunity to accurately locate the service connections during a digital water metering rollout was raised as a benefit. The service connections could then be added into the assets register, **increasing the water utilities asset value**.

*4.3. New Knowledge*

4.3.1. Customer Segments

During the rollout of digital enabled meters, the collection of additional data such as business type (i.e., bakery, legal firm, supermarket, clothing store, school, etc.), effectively an audit of **nonresidential customer property use** by business types, would itself add knowledge. As usage data is collected, the demand profile of that business type is identified. Collated with other similar businesses, patterns emerge and the potential to better forecast demand becomes available.

Digital metering data can help planners and operators to understand **tourist impacts for tourist regions by season and events** [10,75]. Calianno et al. looked at the water use in a tourist mountain area affected by seasonal water demand of the temporary population and irrigation use and noted the lack of water use data at sufficient temporal resolution compared to that in large cities from using smart metering [40]. The increasing water demand in the Mediterranean of tourist resorts that offer golf courses, spas, aquatic parks, swimming pools, and irrigated gardens was the subject of a study by Hof and Schmitt [76]. They urged the use of water metering to gain the necessary water use data for sustainable water demand management.

Many researchers point to the benefit of understanding time-of-day use by residential customer segment [10–12].

### 4.3.2. New Algorithms

Digital water metering data has the potential to enable new insights into the relationship between operating parameters such as total demand, flow rates, weather and meter recording through the application of machine learning, neural networks, cluster analysis, multivariate statistics, artificial intelligence, visualisation, and organisational data literacy [77–79]. From these new mathematical models, algorithms may be developed to improve water utility planning and operations [80]. In the area of customer end-use, Nguyen et al. applied a hybrid combination of the Hidden Markov Model (HMM) and the Dynamic Time Warping algorithm (DTW) to disaggregate household flows [9], Cardell-Oliver applied cluster analysis to identify signature patterns [12], and Fenrick and Getachew advanced statistical methods to measure the impact of frequent billing on demand and to identify long-term appliance efficiency trends [81]. Huang et al. applied dynamic time warping, supervised learning, and calculated the abnormality probability for real-time burst detection in DMAs [82]. Herrera et al. provided an overview of a Special Edition covering technigues hydroinformatics and their application using the realtime data from water sensor devices to a cross section of water supply system problems [83].

Soderberg applied various data analytic methods to water consumption in rental apartments [84]. When collating data from similar businesses, patterns emerged and the potential to better forecast the demand became available. The establishment of **diurnal curves for nonresidential customers by customer type** became a reality. A nonresidential segmentation study [41] that relied on extensive data matching highlighted the benefit of peer comparison even though the base demand data was quarterly readings. A simple scatterplot of a number of schools' usage and student population data provided a visual peer comparison. While most of the scatter points centred around a common line, instances of potential anomalous customer metering could be seen with many, possibly oversized meters (a low consumption for the number of students) and one possible leak (a high consumption for the number of students).

Fernando and Roberts pointed out that, when consumption data was analysed, it enabled micro-segmentation of customers based on consumption patterns [23].

Once the micro-segmentation was established, changes in the consumption pattern at a property might signal a change in business type and trigger a review of the suitability of the current meter.

Digital metering data can help planners **reduce the uncertainty** and use more accurate and efficient models. Experts (E15 and E24) thought the benefit was useful in dealing with the impact of infill development and changing lot sizes. E22 pointed out that the water industry is conservative. E58 raised the issue of planners liking a margin of safety.

Cole and Stewart in 2013 pointed out the need to correctly size customer meters and that the selection could be enhanced by the availability of hourly consumption data [10] and, likewise, that network planners can use hourly consumption data to confirm estimates for "equivalent dwelling"

and "equivalent persons" consumption. A conservative, overestimation of these parameters can result in an oversized infrastructure. The benefit from hourly consumption data extends to wastewater systems where the hourly consumption can be cross-checked with sewer flows and estimates of **inflow and infiltration of the sewer system can be enhanced**. The Experts interviewed (E15, E26, E51, and E59) generally agreed with this benefit. Similarly, hourly consumption can **improve sewer flow modelling**, and volumes can be cross-checked with estimates of inflow and infiltration of the sewer system [11]

Aggregating digital water meter data for multiunit properties can be used to identify **diurnal curves for high-rise building/multiunit properties** unique to different communities [74,85] (E13, E15, E27, E66, and E71). These curves can **inform demand forecasting and revenue projection** [11] and can be used in network models.

Wong and Mui, in a review of water demand models for buildings, identified three model types, deterministic, probabilistic, and demand time-series and proposed a Bayesian approach [86]. After analysing the data from various studies across buildings in Europe, South Africa, and Japan, Wong and Mui concluded that, for maximum simultaneous flow rates, most designs are routinely and substantially oversized to minimize the chance of errors. They then discuss the cost implications for maintenance and energy.

Duncan and Mitchell developed a stochastic demand generator for a simulation modelling of households from high-frequency data logging of appliances [87]. Recent simulation modelling using the results of a high-frequency end-use study of 337 households in the Melbourne metro area [9] and hourly metered data from high-rises in Melbourne led to the development of a meter sizing algorithm and selection application [85]. Testing the new algorithm against meters applied in recent plumbing applications for high-rise buildings showed that the simulated demand was closer to the actual demand than the demand computed through the Plumbing Standard tables and that the meters being applied for were consistently over dimensioned. The interviewed experts (E25, E26, E41, and E58) believed that future data analytics on digital meter data could enable **meter oversizing identifiers** to be developed. Based on end-use modelling, it could be possible to perform a **reverse modelling of household characteristics via demand patterns** and demographic data for the area. Some of the experts interviewed were keen to be able to do this, with E15 and E22 identifying water tank and evaporative coolers as emerging big issues and the potential to better understand residential irrigation use. However E51 felt that privacy controls need to be in place to ensure appropriate confidentiallity of household information is maintained.

Water utilities can learn from data analytics applied to electricty smart meter data [48,88,89]. Similarly, Cominola et al. reviewed the application of data analytics to smart water meter data [90].

### 4.4. Benefit Catalogue—Business Benefits

The abovementioned benefits that water utility businesses can achieve have been catalogued and grouped into categories and subcategories (see Table 2). The changes required to enable the achievement of the benefits are also listed. Further business benefits also attributed to water utilities are included in Table 4 in Section 6; these have been defined as shared benefits.

**Table 2.** The benefits available to water utilities from changing to digital water meters.

| Category | Subcategory | Benefit | Enabler(s)[1] | Reference |
|---|---|---|---|---|
| Operational Cost Savings | Meter Reading | Reduction in Meter Reader charges/Billing Costs | 1 | [1,6,11,13,16,20–22,52] |
| | | Reduction in Special Meter Reads | 1 | [20,21,52] |
| | | Reduction in Estimated Bills | 1 | [20,21,51,91] |
| | | Reduction in Occupational Health and Safety (OHS) incident costs | 1 | [16,22,32,70] |
| | | Reduction in vehicle energy costs (GHG emissions) | 1 | [6,11,20] |
| | Financial Management | Reduction in billing and collection costs monthly billing is coupled with eBilling and direct debit collection | 5 | [1,52] (E17, E26, E38, E43, E58)[2] |
| | | Improved revenue forecasting/recovery | 2 | [11,74,92] |
| | | Cash flow/reduced working capital from Monthly Billing | 5 | [21,54,55] |
| | | Reduce residential nonrevenue water data errors/losses | 1 | [1,22,32,51] |
| | | Reduce nonresidential nonrevenue water data errors/losses | 1 | [1,22,32,51] |
| | | Reduce insurance claim incidents and costs from bursts and leaks | 6 | [56–59] |
| Category | Subcategory | Benefit | Enabler(s)[1] | Reference |
| Operational Cost Savings | Utility Costs | Reduction in wholesale cost of Water | 2 | [21,51,74] |
| | | Reduction in network leaks and other NRW causes (e.g., bursts) | 6 | [6,13,16,21,52,91] |
| | | Better peak water demand management | 2 | [1,13,74] |
| | | Reduction in water pumping cost (GHG emissions) | 2 | [6,52,61,62,74] |
| | | Reduction in water theft | 6, 7 | [6,11,16,51,62,63] |
| | | Reduction in labour costs associated with leak detection | 6, 7 | [1,74] (E17, E36) |
| | Meters | Deferred meter replacement (through water conservation, targeted replacement) | 7 | [11,52] (E17, E25, E26, E31, E41, E70, E72, E80) |
| | Tariffs | More flexible tariffs by industry | 2 | [6,10,16,37,52,70,90] |
| | | Load shifting (levelling) | 2,7 | [16] (E22, E27, E58, E59, E61) |
| | | Improvement in customer service/satisfaction | 3, 4, 5, 6, 8 | [11] [E19, E22, E38, E59, E61] |
| Capital Cost Savings | Planning | Improved Network Planning | 2 | [1,4,10,11,16,52,74,91] |
| | | Deferred network augmentation | 2 | [1,10,11,16,31,36,52,71,74] |
| | Risk | Reduction in Risk premium/Working Capital Costs | 2 | [4,52] (E22, E27, E58, E59) |
| | | Increased value of asset (service connection) | 8 | [36,74] (E19, E59) |
| New Knowledge | Customer Segments | Nonresidential customer property use | 7, 8 | [11,41,74] |
| | | Tourism impacts for tourist region (Seasonal/event) | 7, 8 | [11,40,74] |

| | | | |
|---|---|---|---|
| New algorithms | Understand Time-of-day use by residential customer segment | 7 | [10–12,16,74] |
| | Meter oversizing identifier | 7 | [10,41] (E25, E26, E41, E58) |
| | Reduced uncertainty/reduced risk margin | 2 | [10,74] (E15, E22, E24, E58) |
| | Improved forecasting of sewer flows | 2 | [10] (E15, E26, E51, E59) |
| | Improved demand forecasting and revenue projection | 2 | [11,16,74] |
| | Diurnal curves for nonresidential customers by customer type | 7 | [11,23,41,74,93,94] |
| | Diurnal curves for high-rise building/multiunit properties | 7 | [11,74] (E13, E15, E27, E66, E71) |
| | Reverse modelling of household characteristics via demand pattern | 8 | [74] (E15, E22, E27, E51, E59, E66) |

[1] The enabling change, Automate Meter Reading using AMI, is assumed for all benefits if not listed. [2] Profiles of the Experts are included in Appendix A.

## 5. Customer Benefits

Customers can be expected to benefit directly from water utilities upgrading to digital water metering. Customer benefits could be received in three main categories, namely water usage cost savings, customer service, and new knowledge.

### *5.1. Customer Service*

5.1.1. Usage Cost

With only one water reading and bill every 92 days (i.e., quarter), concealed leaks can waste a lot of water if not noticed. The rising bill may not be acted on immediately, and two or more quarters might pass before the customer reacts. Water companies have developed reporting systems that look at the quarterly reading in an attempt to identify high use [95]. When data analytics have been employed even on quarterly reading data, the time until the customer could be alerted can be much earlier [96].

However, hourly digital water meter readings has provided many water companies with the opportunity to identify anomalous water consumption and to provide **leak alerting to customers** within days of the start of the problem [20,23,97]. The reaction from customers after being advised of a possible leak has been almost always positive and grateful for the **avoidance of high bills** [35]. However, some trials have shown customers do not always get a problem fixed even though they have been advised of a problem [1,36]. Systems such as MiWater provided customers with a leak alert via text message, email, or letter when the system identified that a threshold of nonzero flow over a period of time had been reached. The customer might know the cause of the continuous water use, or it might be a concealed leak or a theft from behind the meter.

Concealed leaks can result in property damage as well as high water bills. Water theft behind the meter also occurs [98]. It follows that if concealed leaks and theft can be detected earlier, property damage and loss would be reduced. Digital water meters and leak alerting can **reduce insurance claims** for the property owner. Claiming on insurance is not easy as it takes effort to make the claim, to provide for access by an assessor, and to argue any rejection by the insurance company, and there is the expense and inconvenience of rectification. Once a claim is made against a policy, the insurance company may add exclusionary clauses [56], deny ongoing coverage, or significantly raise the premium [99,100].

Providing customers with near real-time information on their water use can assist water conservation and lower water bills [5,11–13,15,16,52,90–92]. Liu et al. reported a 4.2% usage **reduction due to customer awareness/education** across an entire sample of households receiving additional

marketing/advertising and reminders for use of a water utility portal in the Mid-Coast Water trial [5]. Such portals might provide an estimation of the next bill based on the trends of usage to date and recent and historical usage patterns. This may lead to a **reduction due to bill prediction** [1,52].

Each of a customer's utilities has their own billing cycle: In Victoria, Australia, it is every two months for gas use and quarterly for water use, the electricity use is billed quarterly with some customers switching to monthly, and municipal rates are payable in ten instalments or as a single lump sum. Budgeting for these payments would be difficult for many customers on low incomes or high expenses relative to income. The WSAA Cost Benefit Analysis spreadsheet [52] included costs and benefits for switching from quarterly to monthly billing but no acknowledgement was made for any water conservation from customers' demand reductions that might result from increased customer awareness of water use habits. As well as reducing consumption, monthly billing would improve customers' ability to budget and improve the utility affordability by reducing the "lumpiness" of bills [101].

The option of monthly billing is now being offered among electricity retailers in Victoria, exploiting the opportunity that smart meters provides for automated reading of meters [55]. Hourly data from digital water meters would also allow monthly billing (or even fortnightly or weekly billing). While many customers might be expected to prefer to remain on quarterly billing, others might switch.

In 2012 Fenrick and Getachew [81], studying the impact of billing frequency, developed a demand function of residential water consumption using data from a 1997 to 2006 panel of 200 Wisconsin water utilities. Unlike previous studies which relied on regression, they assumed a double-log functional form and estimated parameters using a random effects model. Studying water conservation effects of price and other factors, the model was able to evaluate the effect of monthly billing over longer billing periods and the long-term trend in water conservation due to advances in appliance technology. The conclusion reached was positive for the impact of monthly billing on water conservation. They determined a 3.7% reduction in household water usage for monthly billing against a longer billing frequency [81]. Their study confirmed a **reduction in water demand through monthly billing** to the utility through providing more information to their customers and creating an increased awareness of the costs of water. The findings are at odds with the study by Wichman [102] who noted an increase in consumption when the billing frequency for the City of Durham changed from bimonthly to monthly billing but noted the need to rethink the billing information provided to customers.

Monthly billing would require 3 times (or 4 times) the number of bills of quarterly billing and, therefore, 3 (or 4 times) the cost. The cost of shifting to more frequent billing might be offset through eBilling, possibly coupling eBilling with direct debit payment, which might be made mandatory to take-up monthly billing.

In the same study, Fenrick and Getachew also identified the impact of introducing water efficient appliances on total demand and determined a 1.1% reduction in per capita water use due to the household adoption of water efficient technology [81]. Digital water metering may improve the measurement of impact of appliance efficiency on total demand.

### 5.1.2. Complex Property/Multiunit Usage Reconciliation

The Strata Community Association is the peak body representing managers of apartment blocks, unit blocks, retail complexes, commercial, industrial, government, and high-rise buildings. They report that Australia has 270,000 strata developments comprising over 2 million "lots" (properties) [103]. Multiunit property developments comprise check meters on each lot for billing water usage within the lot, main meters for billing of the water usage on common facilities (such as gardens, recreational facilities, and common hot water, etc.), and fire service meters. With quarterly billing Owners Corporations (OC), managers struggle to understand what is happening within the property and conversations with contact centres are described as lengthy and frustrating, and reported water usage-related issues take months to resolve [100].

Armed with evidence from hourly usage graphs across all meters at a property, the OC manager would be able to have a shorter and more informed discussion with the water company. Systems such as MiWater provides for what it calls "private networks" [104], which facilitate access to a web portal for the various submetered points of the overall property.

Digital water metering would enable significant efficiency gains for both the OC manager and for the water company in terms of a **faster and easier reconciliation of bills for properties with multiple accounts** and provide OC managers with the capability to **identify plumbing irregularities in properties with complex plumbing.** In the second structured interview, Utility#13 staff acknowledged that disputes occur more frequently with OC properties and that these disputes are more difficult to resolve as there are more checks required and they involve more people [105]. While one third of the water company's master metered properties already have digital meters, no customer access to the data is provided. Notwithstanding the lack of customer access and self-service, Utility#13 believes that disputes involving digitally metered properties are easier to resolve.

### 5.1.3. New Services

Reading routes are used by meter readers to establish equitable daily work-loads across the billing cycle. The restriction to base the customer's billing day on the reading route that the property is in no longer exists when digital water meters read daily or hourly, every day of the year. The opportunity exists for **customers to nominate their billing day** [52], using a reading taken, say, 14 days earlier. Being able to be billed on a nominated date could improve payment collections as it would fit household budget cycles. Experts interviewed pointed out that offering this service could have a positive effect on paying on-time (E19, E22, and E59) and an increase in customer satisfaction and goodwill (E19 and E27). One expert, E14, pointed to the significant system and cultural changes within the water utility to make it happen.

Evaporative air coolers are considered to have the following advantages as cooling systems: use water as the cooling medium; applicable in hot dry climates (comfort); 100% outdoor air, no recirculation; lower capital cost; and lower operating costs [106]. For these reasons, evaporative coolers are popular in the southern states of Australia. In a trial by Utility#10, 36% of properties had evaporative coolers and, when used on peak days, they accounted for 23% of usage [107]. The water demand is for both evaporative and non-evaporative use by the units. Evaporative demand is between 60 and 100 litres/hour depending on typical seasonal variations in ambient air conditions and model characteristics. Non-evaporative demand (bleed and/or dump) rates of 5 to 30 litres/hour depend on the water quality and settings [108]. While the study reported that they did not encounter units with water losses (i.e., leaks, splash outs, and overflows), it is conceivable that they do exist. Cardell-Oliver described unexpected patterns of water use in a data sample from Kalgoorlie-Boulder and suggests that the possible cause was leaking evaporative coolers when being used [12]. Digital water metering may assist customers in understanding their **evaporative cooler water use** and in identifying excessive water usage based on recorded high hourly water use during operation of a unit caused by poor installation, poor maintenance, a lack of servicing, or mechanical damage.

Residential end-use has been the subject of a number of studies, and from them, new insights have been gained and algorithms have been developed for de-aggregating flows [9]. More complex nonresidential customers with multiple water consuming subprocesses may similarly benefit from high-frequency data logging. From that, logging end-use profiles could be developed and then used for identifying anomalies or inefficiencies [74]. The suggestion that **nonresidential customers might benefit from data logging and analytics** was strongly endorsed by the experts interviewed. E14 saw this as assisting business customers to avoid bill shock and to reduce bills. E15 and E19 saw this as an opportunity to develop many new services to business customers.

### 5.1.4. New Products

**Customising marketing offers** based on consumption patterns can minimize wasted marketing spending [11]. The data analytics expert interviewed suggested, as an example, that targeting high

users might enable marketing offers of washing machines to reduce demand but that it would be "tricky" [74]. Other experts interviewed (E14, E15, E19, E27, and E59) also confirmed this benefit.

When modelling the appliance use within a property, Cominola et al. discussed intrusive metering (one meter per appliance) and nonintrusive metering which relied on algorithms to disaggregate medium resolution (e.g., Hourly) data into possible appliances based on their water use profile [90]. It was suggested that there may be customers, especially large water commercial, agricultural, and industrial users who may wish to have their **data disaggregated by end-use appliance** within a portal and that this could be offered as a new product by water companies. Some of the experts interviewed saw this as a potential new product offer (E26, E27, E40, and E76), many were unsure (E13, E33, and E68), and others saw this for end-use studies only (E22).

### 5.1.5. Security

Customers receive an **improved security** benefit from automated meter reading as the meter reader is not required to enter the property [22].

When Kansas City, USA, implemented an AMI project to rollout digital water meters, it was able to **monitor vacant properties** to ensure that no water was consumed [20]. Their purpose was to bill the owner of the vacant property, if water was used. Looking at the monitoring from the owner's perspective, properties are left vacant from time to time, such as when the owners are away on holidays, it is a rental property between lets, a holiday house between visits, the person living there is in hospital for an extended period, or the owners are selling and have already moved out. Any water use may indicate theft, that squatters have moved in, or that a burst or leak has occurred. For households in Alicante, Spain, users can set alarms for high consumption when the household is vacant or during the night [6]. The potential of vacant property monitoring to assist property owners may not have been considered by water utilities [74]. Setting triggers for leak alerting at very low levels may impose high service costs on the water utility for email or SMS alerts due to the number that could be generated. A fee for service or limiting the alert frequency might be necessary to make the service financially viable for the water utility.

### *5.2. New Knowledge*

### 5.2.1. Appliance Usage/End-Use

Koech and Gyasi-Agyei [13] suggest the continuing development of the information and communication technology will increase the scope of smart water metering in the management of water systems and also enable an integration with appliances within the home. King [109] reported on a number of innovations on the integration between appliances and smart meters. What might have been considered futuristic 10 years ago, the possibility is for the **integration of smart meters with "smart" appliances**.

Appliance efficiency improves in response to product development and differentiation, leading to water saving innovations, and the impact of these savings need to be included in assessing changes in demand and in planning water systems. Fenrick and Getachew, in their study, calculated the annual decrease in water use per household due to the adoption of these water-efficient appliances as 1.1% [81]. It is conceivable that large-scale data collection from digital water meters may be used to monitor this rate of change and the **appliance efficiency impact on total demand** which informs network planning and usage forecasting [110].

### 5.2.2. Benchmarking

If data from digital water metering is combined with other data, benchmarks, i.e., standard operating profiles, may be developed to enable customer comparisons against the benchmarks.

Evaporative coolers make substantial demands of water when in use. The various models and ages have different demand profiles [108,111]. If customers provide details into a web portal of their evaporative coolers among demographic and other appliance details, then this data might be used to develop operating norms as benchmarks against which a customer might check the performance of

their coolers. **Benchmarking water demand of evaporative coolers** has the potential to inform the network planning function and usage forecasting.

Customers can be segmented based on business type or residential property type and further subsegments using coding systems such as Australian New Zealand Standard Industrial Classification (ANZSIC) [93] and Australian Valuation Property Classification Code (AVPCC) [112]. Using the vast amount of data from digital meters, the demand profiles of the segments can become **benchmarks for the customer segments**. Customers might then be able to compare themselves against their peers if presented with simple box-plots of their segment and their consumption [1,41,113].

*5.3. Benefit Catalogue—Customer Benefits*

From the sections above, the benefits that customers can gain have been catalogued and grouped into categories and subcategories. The changes required to enable the achievement of the benefits are also listed. More benefits to customers are included in Table 4 in Section 6, Shared Benefits.

**Table 3.** The benefits available to customers from water utilities changing to digital water meters.

| Category | Subcategory | Benefit | Enabler(s)[1] | Reference |
|---|---|---|---|---|
| Customer Service | Usage Cost | Reduction in cost to customers due to leak alerting | 3 | [11–13,16,20,21,35,54,90,91] |
| | | Reduction due to customer awareness/education | 4 | [5,11–13,15,16,52,90–92] |
| | | Reduction due to bill prediction | 4 | [52,74] |
| | | Reduction due to Monthly Billing | 5 | [20,81,92] |
| | | Reduction in Insurance Claims | 3 | [56,98–100,114] |
| | Complex property/multiunit usage reconciliation | Faster and easier reconciliation of bills for properties with multiple accounts | 3, 4 | [85,100,103] |
| | | Identify plumbing irregularities in properties with complex plumbing | 3, 4 | [100,103] |
| | New Services | Customer selection of Billing Day | 5 | [52] (E14, E15, E19, E22, E27, E59)[2] |
| | | Evaporative cooler water use | 7 | [12,74,106,108] |
| | | Nonresidential customer end-use data logging and analytics | 7 | [74] (E14, E15, E19) |
| | New products | Customised product offers | 8 | [11,74] (E14, E15, E19, E27, E59) |
| | | Disaggregation/Appliance End-use | 7 | [74,90] (E26, E27, E40, E76) |
| | Security | Increased security for home and business owners | 1 | [22] |
| | | Vacant property water use monitoring and alert | 4 | [6,20,74] |
| New Knowledge | Appliance usage/End-Use | Integration of smart meters with "smart" appliances | 7 | [13,74,109] |
| | | Appliance efficiency impact on total demand | 7 | [74,81] |
| | Benchmarking | Benchmarking water demand of evaporative coolers | 8 | [74,108,111] |
| | | Benchmarking customer segments | 8 | [41,74,93] |

[1] The enabling change, Automate Meter Reading using AMI, is assumed for all benefits if not listed. [2] Profiles of the Experts are included in Appendix A.

## 6. Shared Benefits

Some benefits from digital water metering could benefit both the water utility and customers, that is, the benefits of the change is shared between both groups and could also be considered as a win-win. For example, the benefits might deliver better customer service as well as a cost reduction for the business.

*6.1. Customer Interaction*

### 6.1.1. Complaints

Digital water metering can **reduce customer billing complaints** and **improve outcomes from billing disputes**. When Detroit, USA chose the Itron system for meter reading, it believed that it would alleviate large bills for unused water and avoid billing disputes [21]. Kansas City, USA introduced digital metering and enabled call service representative staff and customer access to the water usage data. The project was identified as being responsible for significant drops in the number of field trips generated by billing complaints, improved conversations, and allowed customers to self-service their questions through access to their daily readings [20]. March reported similar outcomes for complaints in Alicante, Spain [6]. In Melbourne, Australia, City West Water reported that inaccurate billing complaints had reduced from 270 (2013) to 1 (2014) after implementing AMI in multilevel developments [1]. A study of 74 high bill complaints at Utility#13 found that the amount in excess of their normal quarterly bill ranged from $24 (25%) to over $6166 (6000%) [115]. A data analytics expert described his experience with data mining the digital meter that supplied data to eliminate bill shock from the digital metering: One month into the quarterly period, he would email the customer if the bill looked like it was $20 over average [74].

Australian State Governments have established independent bodies to assist water utilities, and their customers reach agreement when disputes on various matters cannot be satisfactorily resolved through internal company processes [116–122]. Typically, these disputes are categorised and include billing complaints as well as credit, customer service, provision, supply, land, privacy, and general enquiry. Among the billing dispute issues, Energy and Water Ombudsman of Victoria (EWOV) identified the following sub-issues: High Fees and Charges, Error, Backbill, Estimation, Concession, Refund, Meter, Delay, Format, Tariff, and Other.

It is suggested that if digital water metering can achieve a reduction in customer billing complaints for the sub-issues of High, Error, Estimation, and Meter, then the number of billing complaints that go through to the Ombudsman offices will also be reduced. In Victoria in 2017–2018, these sub-issues amounted to 62% of billing disputes (674 of 1091 cases), which was 35% of all water cases (674 of 1928) [123]. If the number of billing disputes is reduced, it follows that digital water meters will also **reduce the external cost of water Ombudsman services** for billing complaints, and the water companies will benefit from **reduced internal costs of water Ombudsman-escalated complaints** as well as each customer's time and expense.

Case handling within the Ombudsman office requires both their resources and water utility resources [119]. If achieved, reducing the number of cases by 35% could be expected to save considerable resources. The costs for each utility to fund of the Victorian Ombudsman's office is based on their number of customers and the time taken to handle cases [101]. While the fixed charge for the number of customers would not change, the variable charge for case handling should reduce. The Annual Reports of the Ombudsman and water utilities give some clue as to how much money they might be paying to the Ombudsman annually for processing complaints. The internal cost of resourcing the handling of Ombudsman-escalated complaints is unknown.

### 6.1.2. Customer Assistance Programs

Water businesses operate a number of programs that provide customers with assistance when the have concealed leaks or financial difficulty in paying their water bills.

In a Consultant's Corner opinion piece describing the reasons for Detroit Water and Sewerage Department taking up the Itron digital water meter system, the elimination of large bills from unused water is mentioned [21]. DC Water USA was awarded a Utility Professional Best Practices Award based on their development of a High Usage Notification Application for automated notification of high usage at a property [124]. In their paper to the Ozwater Conference in 2017, Boerema et al. reported that, in Sydney, Australia, concealed leaks cost customers over $5M in 2015–2016 with over 3400 customers affected at an average of $1518 worth of water they could not have known was leaking [95]. In the paper, they state that these leaks were best identified via frequent monitoring and they developed an algorithmic method of detection based on quarterly bills in the absence of smart metering solutions. They further pointed out that the period that a concealed leak runs when quarterly billing is the only trigger for action by the customer can be multiple billing cycles. At City West Water, 56% of High Usage Leak Allowances (HULAs) were being detected by billing period exception reporting. Roberts and Eng developed an algorithm, CheckMate-Lite, to identify HULAs using only quarterly readings and achieved a 78% detection rate [96].

The *Beaudesert Times* reported a case of bill shock from a $500 water bill from a pipe burst and that Queensland Urban Utilities (QUU) had reimbursed the property owner [125]. In the article, it reports that a QUU spokeswoman said "Mr Costello's case was not uncommon as the organisation received about a thousand of similar cases every year". In the same article, QUU's leak insurance product, Concealed Leak Insurance, is mentioned along with the leak detection device and switch, AquaTrip. QUU charge $16 pa for leak insurance for excess water usage to a maximum of $10,000 [126].

The extent of the problem faced by both water businesses and customers is highlighted in annual reports. For example, City West Water, a metropolitan water utility of approximately 460,000 customers (2018) reported in their Annual Reports [127–129] that they may waive part of the customer's debt when there has been a significant undetected leak on a customer's property, unexplained high usage on a customer's account, or in the case of financial hardship. In the four financial years 2015, 2016, 2017, and 2018, this amounted to write-offs of $984,000, $1,056,000, $754,000, and $567,000 (AUD$), respectively. These figures exclude the internal staff costs of processing these HULAs. Reducing the period of the concealed leak from months to days has the potential to **reduce the HULA write-off**. If 100 days can be reduced to 5 days, significant savings are possible.

While water utilities have service restriction procedures and debt recovery processes, they also have hardships and other policies in place to assist customers with high bills. These include assistance programs such as the Victorian Government's Utility Relief Grant Scheme [130,131], the New South Wales (NSW) Payment Assistance Scheme [132], their internal water utility's own hardship schemes [45,133], and payment plans [134]. While assistance programs are generally for customers willing to pay but lacking the capacity to pay, there is the potential that concealed leak identification and other digital water metering initiatives may benefit both customers and water utilities through both **reduced plumbing assistance cost** and **reduced Government Assistance Grants**, by allowing at-risk customers to be identified much earlier and for budgets to be spent more broadly or reduced. The experts interviewed (E17, E20, E33, E59, and E82) generally agreed that an early detection should lead to an early intervention and less credit problems and, so, should reduce the assistance program costs. However, some experts (E27 and E35) pointed out that more people will be alerted to small and medium leaks and seek assistance for these.

### 6.1.3. Credit Management

Water utilities also take action when the request for payment is not honoured. These actions include restricting the flow rate at the property and initiating legal action. If high bills and billing disputes can be reduced through leak alerting, customer access to medium resolution usage data, and monthly billing, the outcome may be **reduced supply-restriction-case costs** and **reduced debt-recovery/legal-action-case costs** [45]. The experts interviewed agreed, believing that there would be fewer credit management cases (E22) and fewer estimated readings; the customers would be more

likely to pay (E63); and that digital water meters would provide a chain of evidence for the water utility leaving customers with much less legal recourse (E27 and E59).

### 6.1.4. Customer Interactions

With more data from digital meters and possibly introducing monthly bills instead of (or as well as) quarterly, many of the experts interviewed felt that more calls, not fewer, would be generated, at least initially. However, the experience of the District of Columbia Water & Sewer Authority in the USA, that was working with IBM on new systems, reported achieving a **reduction in billing-related calls** after introducing automated meter readers [24]. Likewise, Kansas City noted that 2000 unique customer visits per month to its web portal took pressure off the call centre [20]. WSAA included line items in its sample cost–benefit analysis spreadsheet covering calls with respect to billing and an estimated call rate of reduction [52].

Sonderlund et al. [14] provided a review of consumption feedback strategies to households. A study in Israel identified the priority interests of customers for abnormal usage [8]. MidCoast Water's study in Australia, titled *Home Water Update*, looked at customer preferences for **enhanced communication** using data from smart water metering [71]. Data presentations including usage data (i.e., volumes, self-comparisons, and normative comparisons involving own street or suburb) and recommendations for savings were found to be preferred. Water utilities indicated that customer self-service access to the data presentations was preferred [1].

### 6.1.5. Goodwill

Once established, the remote detection and notification of concealed leaks is a low-cost and effective way to support many customers in a stressful time of need that often leads to indebtedness [95]. In the Wallis Consulting Group report to the Victoria's Essential Services Commission, they reported that a proactive attitude adopted by the water utility was highly valued by customers and included a typical comment made by a surveyed customer: "Make the bills a little more frequent so they're easier to budget for. Also make the bills simpler to read. All the usage stuff is confusing." [134]. The experts interviewed expressed similar views on customers appreciating water utilities being proactive. E68 went further and talked about the need to extend leak alerting with good processes, water efficiency, and vulnerable customer programs to deliver a holistic approach to avoid what would otherwise be a very negative experience.

Four distinct sources of goodwill benefits to the water utility are suggested. When alerted to leaks at their property, customers are grateful [35], an example of the **improvement of value of goodwill from information sharing** that water utilities might enjoy (E17, E25, E41, E49, and E80). Collecting 2200 consumption readings per meter per quarter enables new and innovative products and services to be developed. When customers take up new products and services, water utilities could receive an **improvement of value of goodwill from new products and services**. That same data when used for optimised planning and operation that saves the water utility money may result in an **improvement of value of goodwill from customer recognition of operational efficiency and capital management**. With a consumption for each hour of each day for each customer being recorded, new tariffs might be developed that shift the load and reduce operating costs for the water utility but also reduce costs for the customer. This facilitated win-win situation might result in an **improvement of value of goodwill from more flexible tariffs**.

The experts interviewed (E14, E17, E35, E64, and E81) were positive towards a more flexible product and services range that treated customers more individually. E24 felt that it was important that there was some consistency of products across water utilities. E15, E35, and E64 all felt that customers expect water utilities to be efficient and will not reward them with improved goodwill. Other experts were positive that customers would recognize the effort of water utilities to be more efficient but that the outcomes need to be socialized and reflected in their water bills. The potential for increased goodwill from more flexible tariffs received lukewarm responses from the experts interviewed. E35 and E40 stressed the need for transparency, while E22's opinion was that customers were anti-Time of Use (TOU) tariff following the "electricity debacle".

Goodwill may not appear in the balance sheet of Government-owned and Council-owned water utilities but could reveal itself in shorter, less confrontational interactions between staff and customers, a greater trust in water utility messages, and more successful marketing initiatives [30]. Prevos [135] goes further and notes that customers have a low level of involvement with their water utility and that a strong relationship where customers have a positive image of the utility are more likely to be accepting of unavoidable emergencies. Happier staff would result in having lower turnover, thereby saving recruitment costs and attracting better candidates [136]. Customer satisfaction surveys have shown improvements following digital metering and related business changes [1,23,137].

### 6.2. Regulation/Compliance

### 6.2.1. Metering

Australia's National Metering Institute (NMI) requires water businesses to use water meters that measure water usage within nominated tolerances [65]. As nonresidential customers can represent a large proportion of total customer demand, the selection of the meter is important as low flow rates might be undetected or under-detected, resulting in nonrevenue water, and high flow rates can damage the meter or cause excessive wear [85]. While plumbing codes provide tables to guide hydraulic engineers when selecting a pipe size for the main meter and, therefore, the size of the main meter [138], Cole and Stewart pointed out the benefit of hourly consumption data and that meter sizing and selection should be based on diurnal patterns [10]. Customer segmentation, usage profiling from digital meter data, and short interval data logging could assist in building models to **improve meter sizing for nonresidential customers**.

Candeliari offers methods to detect metering faults and other anomalies [62]. Li et al. in their IEEE paper on usage analysis, nominated a number of metering benefits in the area of meter anomaly detection. These included **meter failure analytics** with external factor correlation, leading to a **tighter meter performance/NMI Compliance monitoring and meter silting detection in large meters** for which they claim a success rate during testing of 86%. Along a similar line, Beal and Flynn identified **detecting revenue losses caused by a declining or failed meter accuracy after break in main** as a benefit [1]. E81 viewed meter compliance as a significant benefit of digital metering but, like others, was unsure how this could be done. E72 identified using historical reads of the meter to test that meter reads remained within tolerance. E17 thought that digital metering might assist customers in having faith in their meter and reducing queries ending in meter removal and testing.

### 6.2.2. Monitoring

When water restrictions are imposed on communities, the intention is for the "pain" to be shared equitably across the community. **Monitoring consumption compliance with restrictions and regulations** is significantly enabled by digital metering that records hourly consumption.

Koech et al. discussed the occurrence of droughts and water restrictions in Australia. They indicated that water metering has a role to play in monitoring if consumers are adhering to water usage regulations [13]. Cardell-Oliver described water use signature patterns and the potential for their use for water conservation in Western Australia where there are periods during the year when garden watering systems are to be switched off [12]. Other authors described opportunities to automate the monitoring of water use when restrictions such as "odds and evens" are in place [10,16,90] and to reduce the cost of monitoring [52]. Some authors was further and put digital water meter into the context of an intelligent urban water network (IUWN) where not only volume but also water quality, pressure, and asset condition were measured [16,17]. Mutchek and Williams [17] described the establishment of smart grids in Singapore and San Francisco Bay area. Similarly, Yarra Valley Water in Victoria successfully trialled the Hydrant Insertion Device in Craigieburn and the Ridge-Monbulk areas for monitoring a range of water quality and pressure as part of an intelligent network [38]. A **reduction in audits required (targeted through smart water meter (SWM) water**

**quality testing)** might be achieved by integrating water quality testing into the digital metering data collection [52].

*6.3. Benefit Catalogue—Shared Benefits*

From the sections above, the benefits that both customers and water utility businesses can gain have been catalogued and grouped into categories and subcategories. The changes required to enable the achievement of the benefits are also listed.

**Table 4.** The shared benefits available to both customers and water utilities from digital meters.

| Category | Subcategory | Benefit | Enabler(s)[1] | Reference |
|---|---|---|---|---|
| Customer Interaction | Complaints | Reduced Customer Billing Complaints | 3, 4, 5 | [1,6,20–22,92] |
| | | Reduced external cost of Ombudsman referred complaints | 3, 4, 5 | [117–123] |
| | | Reduced internal costs of Ombudsman referred complaints | 3, 4, 5 | [45,117–123] |
| | | Improved outcomes from billing disputes | 3, 4, 5 | [20,21,92] |
| | Customer Assistance Programs | Reduced HULA (High Usage Leak Allowance) costs from concealed leaks | 3, 4, 5 | [21,70,95,96,124,125] |
| | | Reduced plumbing assistance cost | 3, 4, 5 | [134] (E17, E20, E27, E33, E35, E59, E82)[2] |
| | | Reduced Government Assistance Grants | 3, 4, 5 | [130,132,134] |
| | Credit Management | Reduced supply restriction case costs | 3, 4, 5 | [45] (E22, E27, E33, E59, E63) |
| | | Reduced debt recovery/legal action case costs | 3, 4, 5 | [45] (E22, E27, E33, E59, E63) |
| | Customer Interactions | Reduction in Call Centre Calls | 4 | [1,20,24,52] |
| | | Enhanced communications | 4 | [1,8,71] |
| | Goodwill | Improvement of value of goodwill from information sharing | 3, 4, 5 | [1,23,35,136] (E17, E25, E41, E49, E80) |
| | | Improvement of value of goodwill from new products and services | 4, 8 | [23] (E14, E17, E35, E64, E81) |
| | | Improvement of value of goodwill from customer recognition of operational efficiency and capital management | 2, 4, 6 | [35,36] (E14, E15, E35, E49, E64, E80, E81) |
| | | Improvement of value of goodwill from more flexible tariffs | 3, 4, 5, 6, 8 | [11,20,92] |
| Regulation/ Compliance | Metering | Improved meter sizing for nonresidential customers | 7 | [10,41,85,86] |
| | | Tighter meter performance/National Metering Institute (NMI) Compliance monitoring | 7 | [1,11] (E17, E27, E33, E72, E81) |
| | | Meter failure analytics | 7 | [1,11,41,74] |
| | | Meter silting detection (large meters) | 7 | [11,74] |
| | | Detect revenue losses caused by declining or failed meter accuracy after break in main | 7 | [1,11,74] |
| | Monitoring | Automated regulation compliance monitoring | 7 | [10,12,13,16,52,74,90] |

| | | |
|---|---|---|
| Water Quality: Reduction in audits required (targeted through SWM water quality testing) | 7 | [16,17,38,52] |

[1] The enabling change, Automate Meter Reading using AMI, is assumed for all benefits if not listed. [2] Profiles of the Experts are included in Appendix A.

## 7. Benefits Taxonomy and Enablers

The benefits catalogue tables in Sections 4.4, 5.3, and 6.3 above are shown diagrammatically (Figure 2). The chart illustrates a taxonomy for the benefits of digital water metering within which water utilities could look to find benefits specific to their organization and regulatory environment.

To achieve these stated benefits, the water businesses would be required to make one or more changes to their systems, processes, and resources (refer Section 3 above). The graph (Figure 3) shows the number of benefits that would be impacted by each of the enabling changes. From a water utility business perspective, the green and light blue bars represent potential direct benefits to their business (cost savings or income improvements). The yellow and light blue bars would contribute to a return to the business in the form of cost savings through goodwill and an improved customer satisfaction score.

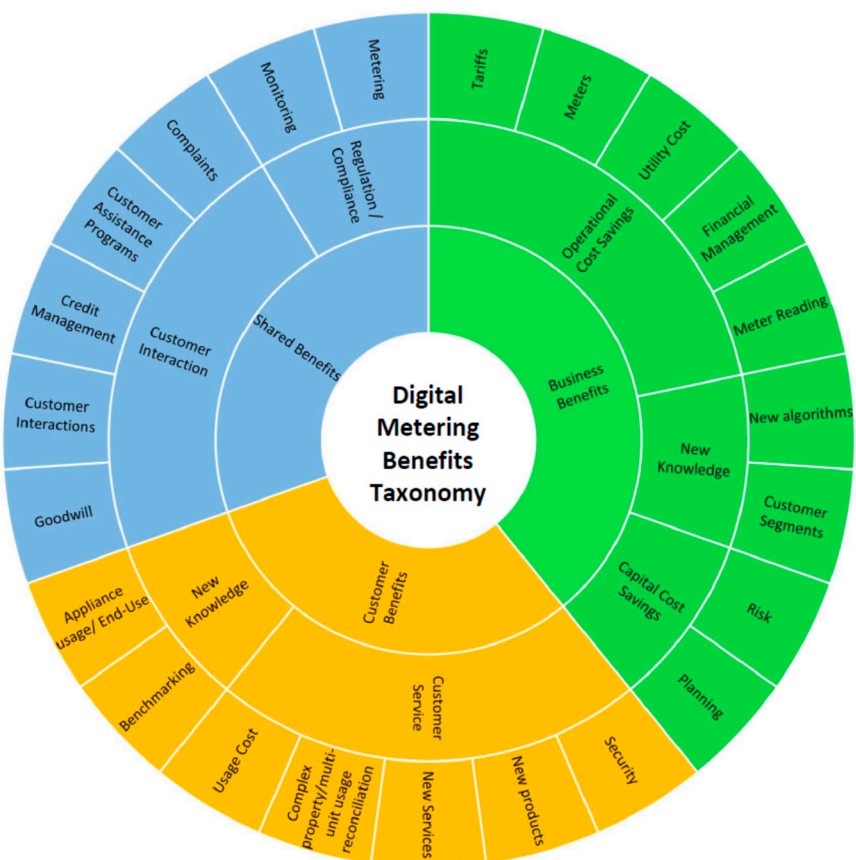

**Figure 2.** The taxonomy of digital meter benefits by beneficiary group.

While many benefits will lend themselves to quantification studies, others will not. Some alternate approaches may be required. A technique for quantifying benefits considered as less-tangible or intangible would further assist project planners.

The further development and extension of Digital Metering Cost Benefit Analysis spreadsheets such as the WSAA spreadsheet to include these new and additional benefits would be useful to the industry to prompt the consideration of other potential benefits. The enabling changes might also be included to prompt consideration of the extra costs required to change systems, processes, and resources in order to achieve the benefits.

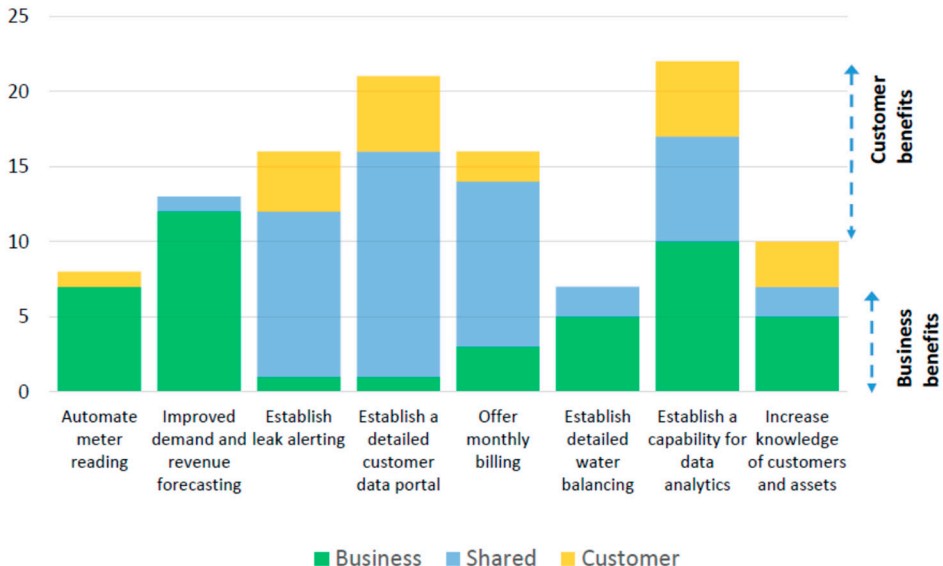

**Figure 3.** The number of benefits that each change impacts[1]. ([1] Enabler#1 (Automate meter reading using AMI) impacts all seventy-five benefits. Only the eight benefits where it is the only enabler required are shown.).

## 8. Conclusion

This study identified seventy-five benefits of digital water metering. Of these, fifty-seven were potential benefits to the water utilities and forty were of potential benefit to customers, noting that twenty-two benefits were common to both customers and water businesses. Many of these benefits might be considered previously unreported. The purpose of the study was to compile the list and to validate them through expert opinion. No attempt has been made to quantify the potential value that they may bring to a business case at this stage. With the research team being located in Australia, many of the industry literature sources have been Australian water utilities and the structured interviews were conducted with Australian experts. While it could be expected that many benefits would be valid in advanced economies internationally, these additional benefits should also be confirmed in other countries. However, simply changing or upgrading mechanical meters to digital would not yield many of these potential benefits. Eight system, process, or resource changes were identified and are listed in this study. The changes required to achieve the benefits were also listed against each benefit. This study has neither attempted to cost these changes nor to undertake a cost benefit analysis at this stage, but this is the intended future research of the authors.

The taxonomy developed provides a framework in which water utilities might focus attention to identify those benefits that are relevant to their operation and customers.

It is hoped that this study provides some insight and further evidence of the potential digital water metering benefits such that water utilities can better frame their business cases both for senior management approval and for persuading customers of the merits of upgrading to digital water meters.

**Author Contributions:** Conceptualization, I.M., R.A.S., O.S. and R.K.; Methodology, I.M., R.A.S., O.S. and R.K.; Formal Analysis, I.M.; Data Curation, I.M.; Writing—Original Draft Preparation, I.M.; Writing—Review and Editing, I.M. R.A.S. and O.S.

**Funding:** The work was completed as part of a higher degree by research program offered by Griffith University. No funding has been provided to support the research reported in this paper.

**Conflicts of Interest:** The authors declare no conflicts of interest.

## Appendix A. Profile of the Experts Interviewed

Fifty-Two industry experts participated in structured interviews seeking their opinions of the list of possible benefits. Their experience and expertise is recorded in Table A1.

**Table A1.** The business area experience of industry experts.

| Expert | a[1] | b | c | d | e | f | g | h | i | j | k |
|---|---|---|---|---|---|---|---|---|---|---|---|
| E13[2] | | | | | | | | | | | Water Efficiency |
| E14 | | | | | | | | | | • | R&D |
| E15 | • | • | • | • | | | • | | • | | |
| E17 | • | | | • | | | | | | | |
| E18 | | | | | | | • | | | | |
| E19 | • | • | | | | | | | | | |
| E20 | • | | | | | | | | | | |
| E22 | | | | • | | | | | | | Water Efficiency, Asset Management |
| E24 | | • | • | | • | • | • | | | | Integrated Water Management |
| E25 | | • | | | | | | | | | |
| E26 | | | | | | | | | | | Management |
| E27 | • | • | • | • | | | | • | • | | |
| E31 | • | | | • | | | | | | | Planning and Maintenance |
| E32 | • | | | | | | | | | | |
| E33 | • | | | • | | | • | | | | |
| E35 | • | | | | | • | | | | | |
| E36 | | • | • | | | | | | | | |
| E37 | • | • | | | | | | | | | |
| E38 | | | | | • | | | | | | |
| E39 | • | | | • | | • | | | | | |
| E40 | • | | | | | | | | | | |
| E41 | | | | • | | | | | | | |
| E43 | • | | | • | | | | | | | |
| E45 | • | | | | | | | | | | Water Efficiency |
| E47 | • | | | • | | | | | | | |
| E48 | | | | | | | • | | | | |
| E49 | • | | | | | | | | | | |
| E51 | • | | | | | | • | | | | Data Science |
| E55 | • | | | • | | | | | | | |
| E58 | • | • | • | • | • | | • | • | | | |
| E59 | • | • | • | • | | | • | | • | | |
| E61 | | | | | • | | | | | | |
| E63 | • | | | | | • | | | | | |
| E64 | • | | | | | | | | | | |
| E65 | • | | | | | | | | | | |
| E66 | | • | | | | | | | | | |
| E68 | • | | | | | | | | | | Water Efficiency |
| E70 | • | | | • | | | | | | | |
| E71 | | • | • | • | | | | | | | |
| E72 | • | | | • | | | • | | | | Contract/Stakeholder Management |
| E73 | • | | | | | | | | | | Communications and Digital |
| E74 | • | • | | • | | | | | | | |
| E76 | • | | | | | | | | | | |
| E79 | | • | • | | | | | | | | |

| Expert | a[1] | b | c | d | e | f | g | h | i | j | k |
|--------|----|---|---|---|---|---|---|---|---|---|---|
| E80 |  | • |  |  |  |  |  |  |  |  |  |
| E81 | • |  |  | • |  |  |  |  |  |  |  |
| E82 | • |  |  |  |  |  | • |  |  |  |  |
| E86 | • | • |  | • | • |  | • |  |  |  |  |
| E89 | • | • | • | • | • | • | • |  |  |  |  |
| E90 | • |  |  | • |  |  |  |  |  |  |  |
| E91 |  | • | • |  |  |  |  |  |  |  |  |
| E92 | • |  |  | • |  |  |  |  |  |  |  |

[1] Legend: Business area experience of industry experts: a, Customer Service; b, Engineering Planning; c, Engineering Operations; d, Metering; e, Finance; f, Legal/Regulation/Corporate Services; g, Senior Management; h, Human Resources; i, Information Technology; j, Academic; k, Other. [2] The expert identification numbering starts at 13 due to the development and testing of the interview questions. Gaps in the numbering were created when participants started but did not complete the interview.

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
