# Peer review of "Revealing Unreported Benefits of Digital Water Metering: Literature Review and Expert Opinions"

_water, doi:10.3390/w11040838_

Round 1
Reviewer 1 Report
The subject of this paper is interesting for people interested in planning and operation of water supply systems.
This paper is about Digital Water Metering (Automated Meter Reading – AMR, and Advanced Metering Infrastructure - AMI) and the potential benefits for both water utilities and customers.
The paper presents an extensive literature review (journal and conference papers describing case studies and reviews of digital metering deployments and trials, and annual reports from water utilities), and interviews of industry experts.
The paper is well structured and the text is well written.
Figures provide important additional information and tables help in synthetizing the information presented.
The paper presents many references, from older to very recent ones.
Being so, my opinion is that the paper should be published as is.
Author Response
Revisions to MDPI-Water Paper – water-471136
Revealing Unreported Benefits of Digital Water Metering: Literature Review and Expert Opinions
Revision 1 (R1) – April 2019
Three (3) Reviewers; R1: Accept in present form; R2: Major revisions; R3: Minor revisions.
We offer the following responses to the reviewer comments.
Reviewer 1 – Accept in present form.
Comment 1.1: The subject of this paper is interesting for people interested in planning and operation of water supply systems. This paper is about Digital Water Metering (Automated Meter Reading – AMR, and Advanced Metering Infrastructure - AMI) and the potential benefits for both water utilities and customers. The paper presents an extensive literature review (journal and conference papers describing case studies and reviews of digital metering deployments and trials, and annual reports from water utilities), and interviews of industry experts. The paper is well structured and the text is well written. Figures provide important additional information and tables help in synthetizing the information presented. The paper presents many references, from older to very recent ones. Being so, my opinion is that the paper should be published as is.
Response 1.1: We thank reviewer 1 for their positive feedback and accepting the paper in its present form. We have revised the paper to polish the English language and addressed comments of reviewers 2 and 3. Overall, with this paper, we have explored the potential for benefits across all water utility business areas and to customers. Through this broad consideration and cataloging we hope that staff outside just planning and operating the water networks will look to ways to exploit digital metering to improve efficiency, work smarter, develop new products and services and positively assist the broad range of customers of water utilities.

Reviewer 2 Report
While the value drivers of smart metering have been extensively and correctly identified , the reasons why the roll out of city metering is so slow have not been really explored.
These might be social reasons which might not be so relevant in Australia or other countries like US/UK but are very strong in continental europe where water is still largely perceives as something public whose pricing revenue models obey different rules .
Moreover how should a cost/benefit analysis be performed of a city metering project has no been really even attempted, let alone performed. Indeed, in a journal grade paper one is expecting a methodologically relevant proposal to be put forward and motivated, not just a long list of benefits.
It should be explained which profitability indexes should be considered and how the computations should be set up and meaningfully interpreted. Are there public contributions to be factored in the analysis? How the deferred network augmentation is factored in the analyis: might an option framework be useful?
Improved demand forecasting should be dealt with in a properly general way with reference to “state of the art “ contributions.
The issues of monitoring water quality and pressure and asset management should be linked to the availability of new data in a scientifically argued way.
The issue of resilience is never considered which is very important also from the point of view of service level.
Data analytics is seen as just a reporting tool, neither mention is made nor awareness is shown of the use of advanced machine learning techniques.
Last but not least the paper deals with the Australian situation only.
I consider the paper, in its present form, unfit for publication in a scientific journal: it needs extensive rewriting as to offer a coherent vision of what smart water and metering in particular will ring forward for the water industry and all its stakeholders.
The analysis of the technologies , including those related to data acquisition and processing, should be done as it befits a scientific journal.
Author Response
Revisions to MDPI-Water Paper – water-471136
Revealing Unreported Benefits of Digital Water Metering: Literature Review and Expert Opinions
Revision 1 (R1) – April 2019
Three (3) Reviewers; R1: Accept in present form; R2: Major revisions; R3: Minor revisions.
We offer the following responses to the reviewer comments.
Reviewer 2 – Major revisions.
We thank reviewer 2 for their comments and offer the following responses to individual issues raised.
Comment 2.1: While the value drivers of smart metering have been extensively and correctly identified, the reasons why the roll out of city metering is so slow have not been really explored. These might be social reasons, which might not be so relevant in Australia or other countries like US/UK but are very strong in continental Europe where water is still largely perceives as something public whose pricing revenue models obey different rules.
Response 2.1:
Examining the barriers or impediments of digital metering diffusion rates and success is a separate study that is considered outside the scope of the present study. In the revised R1 manuscript, we have revised the introduction section to include sentences to briefly explain that there are presently a number of barriers to obtaining these potential benefits of digital metering programs (see lines 37-44).
Comment 2.2: Moreover how a cost/benefit analysis should be performed of a city metering project has no been really even attempted, let alone performed. Indeed, in a journal grade paper one is expecting a methodologically relevant proposal to be put forward and motivated, not just a long list of benefits.
Response 2.2: The present study provides a comprehensive review of all benefits and includes those included in the literature, industry reports as well as those identified from structured interviews with numerous industry professionals. As mentioned, only about half of the mentioned benefits are currently reported in the literature, making this taxonomy of benefits a valuable contribution. The research team seek to compile evidence and valuation techniques for the complete list of benefits as well as case studies for utilities in future work; however, this is a long-term significant program of research works, and the findings will be covered in a number of research papers. At this stage, it would not be possible to include a whole research program within one research publication. We have further expanded the future work section to explain this future program of works(lines 975-976).
Comment 2.3: It should be explained which profitability indexes should be considered and how the computations should be set up and meaningfully interpreted. Are there public contributions to be factored in the analysis? How the deferred network augmentation is factored in the analysis: might an option framework be useful? Improved demand forecasting should be dealt with in a properly general way with reference to “state of the art“ contributions.
Response 2.3: We appreciate the reviewers comment on the many issues and opportunities related to the digital transformation of the water industry, including, but not limited to: (1) profitability indices; (2) new demand forecasting techniques; (3) sensors for managing intelligent water grids for water quality and pressure; and (4) water network resilience. We would like to investigate all of these issues and opportunities, as they are significant. However, all of these aspects related to digital metering require a dedicated research investigation and dedicated research papers to describe the associated studies. The authors feel that the current paper is already quite long at 15,000 words and would become disjointed if it addressed all of these gaps in the research in a robust methodological way.
Saying that, we have tried to highlight some of these aspects better in the revised R1 manuscript. In line 50-51 and 70-71 we have better explained that costing and cost benefit analysis methods is beyond the intended scope of the paper. Sentences added (lines 329-330, 345, 469-471, 914-919, 917-922). The other issues raised are outside the intended scope of the paper.
Comment 2.4: The issues of monitoring water quality and pressure and asset management should be linked to the availability of new data in a scientifically argued way.
Response 2.4: The issue of monitoring water quality and pressure and asset management is covered in lines 912-914. In lines 914-917 we have been added evidence to use of smart meters for monitoring these issues.
Comment 2.5: The issue of resilience is never considered which is very important also from the point of view of service level.
Response 2.5: A sentence has been added (lines 41-42) into the Introduction referencing Mutchek and Williams’ 2014 paper to specifically address the issues of sustainability and resilience of water utilities. The California Water Association’s urging to implement AMI in the face of the crippling drought is included and referenced (lines 42-44).
Comment 2.6: Data analytics is seen as just a reporting tool, neither mention is made nor awareness is shown of the use of advanced machine learning techniques.
Response 2.6: The paper devotes many lines to the area of new algorithms (Lines 457-526). We have added a list of advanced data analytic techniques and reference papers by Gartner (lines 457-461).
We have provided examples of the application of advanced analytic techniques to various issues of data analytics of digital water data (lines 463-474). The paper from Li et al is referenced extensively throughout the paper. Enabler#2 nominates advanced data analytics without specifying any particular technique (line 158). Enabler#7 (lines 219-230) also mentions “the employment of trained statisticians and data scientists”. “Big data and advanced analytics” is mentioned. And lines 228-230 have been reworded and references added. Lines 380-382 added to reference a simple statistical technique used for effective failing meter detection. Candelieri’s work has been included and has been referenced (lines 329-330, 345, 894). The overview by Herrera et al for the MDPI-Water Special Edition on Advanced Hydroinformatic Techniques for the Simulation and Analysis of Water Supply and Distribution Systems is mentioned and referenced (line 468).
Comment 2.7: Last but not least the paper deals with the Australian situation only.
Response 2.7: The literature review is international and covers a range of benefits mentioned from studies across the globe. Due to the location of the research team, the structured interviews were conducted with Australian experts. However, we have acknowledged that while we believe that many of the mentioned benefits from industry practitioners in Australia would also be benefits in other advanced economies, we have added an additional sentence in the Conclusion section (lines 957-961) to indicate that these additional benefits should be confirmed in other countries.
Comment 2.8: I consider the paper, in its present form, unfit for publication in a scientific journal: it needs extensive rewriting as to offer a coherent vision of what smart water and metering in particular will ring forward for the water industry and all its stakeholders. The analysis of the technologies, including those related to data acquisition and processing, should be done as it befits a scientific journal.
Response 2.8: We acknowledge that all the gaps in research on digital metering mentioned by the reviewer are important gaps to be addressed by researchers. However, we feel that all these gaps cannot be covered within one paper. Moreover, the current paper is already long in order to explain the literature, methods and results for the present study (i.e. over 30 pages and over 15,000 words). The scope of works in the study has been purposely focused on building a taxonomy of benefits of digital metering. Presently, researchers and industry do not have such a comprehensive taxonomy that can help them to undertake a complete cost-benefit analysis of digital metering programs. We acknowledge that there is much more work to do to collect existing reported evidence for each currently known benefit and to also create valuation methods for the previously unknown benefits, analyse the various technological solutions, etc. However, to include such work as well as other requested work into this paper would make the current paper too long and disjointed.

Reviewer 3 Report
The paper is a good review of the benefits of Digital Water Metering.
I agree with the subdivision of paragraphs related with the different topics.
It is important the author furnish some indications about a weight that each benefit give for the user, the management or the industry.
I hope the authors afford in a successive paper also a simulation in which define one ore more functions based on some of the parameters and they compare the results of each function to understand the importance of the single parameter.
Author Response
Revisions to MDPI-Water Paper – water-471136
Revealing Unreported Benefits of Digital Water Metering: Literature Review and Expert Opinions
Revision 1 (R1) – April 2019
Three (3) Reviewers; R1: Accept in present form; R2: Major revisions; R3: Minor revisions.
We offer the following responses to the reviewer comments.
Reviewer 3 – Minor revisions.
We thank reviewer 3 for their feedback and offer the following responses to individual issues raised.
Comment 3.1: “It is important the author furnish some indications about a weight that each benefit give for the user, the management or the industry.”
Response 3.1: The contribution of each benefit to the business case of a digital metering project is highly specific to its particular situational context (i.e. city size and density, technology used, etc.). We do acknowledge that some of the benefits identified will contribute more than others to the overall return on investment of the capital investment.
Future work of the researchers intends to value the various benefits and work out the total return on investment. However, this research objective was beyond the scope of the present paper which was focused on creating the benefits taxonomy. An improved clarification of the scope of the present paper has been provided in the revision (Lines 50-51 and lines 966-968.). The weight (benefit contribution) of each benefit would depend on the situation of each water company. The difficulty in quantifying benefits is discussed in lines 20-21 and 946-948..
Comment 3.2: I hope the authors afford in a successive paper also a simulation in which define one or more functions based on some of the parameters and they compare the results of each function to understand the importance of the single parameter.
Response 3.2: The authors intend to undertake successive papers. This future work will be focused on valuation methods and the various parameters that contribute to these benefits. The need for this work is discussed in lines 947-948 and 975-976.

Round 2
Reviewer 2 Report
The paper can be now accepted for publication